# Challenges for Better Diagnosis and Management of Pancreatic and Biliary Tract Cancers Focusing on Blood Biomarkers: A Systematic Review

**DOI:** 10.3390/cancers13164220

**Published:** 2021-08-23

**Authors:** Hiroto Tominaga, Juntaro Matsuzaki, Chihiro Oikawa, Kensho Toyoshima, Haruki Manabe, Eriko Ozawa, Atsushi Shimamura, Riko Yokoyama, Yusuke Serizawa, Takahiro Ochiya, Yoshimasa Saito

**Affiliations:** 1Division of Pharmacotherapeutics, Keio University Faculty of Pharmacy, 1-5-30 Shibakoen, Minato-ku, Tokyo 105-8512, Japan; tominagahiroto@keio.jp (H.T.); 0808oocc@keio.jp (C.O.); toyoken724@keio.jp (K.T.); haruki.may19@keio.jp (H.M.); eriko.ozw@gmail.com (E.O.); btw.orz.321427@keio.jp (A.S.); rikoy@keio.jp (R.Y.); serizawa.y29@keio.jp (Y.S.); 2Department of Molecular and Cellular Medicine, Institute of Medical Science, Tokyo Medical University, 6-7-1 Nishishinjuku, Shinjuku-ku, Tokyo 160-0023, Japan; tochiya@tokyo-med.ac.jp

**Keywords:** biomarker, chemoresistance, liquid biopsy, microRNA, long non-coding RNA

## Abstract

**Simple Summary:**

Pancreatic and biliary tract cancers are malignant tumors that have a very poor prognosis and are resistant to chemotherapy. The later a cancer is detected, the worse the prognosis becomes; therefore, early detection is important. Biomarkers are physiological indices that serve as a guide to indicate the presence or absence of a certain disease, or its progression. The purpose of our research is to summarize previously reported biomarkers for the diagnosis and prognosis of pancreatic and biliary tract cancers.

**Abstract:**

Background: pancreatic cancer (PCa) and biliary tract cancer (BTC) are cancers with a poor prognosis and few effective treatments. One of the reasons for this is late detection. Many researchers are tackling to develop non-invasive biomarkers for cancer, but few are specific for PCa or BTC. In addition, genetic abnormalities occur in cancer tissues, which ultimately affect the expression of various molecules. Therefore, it is important to identify molecules that are altered in PCa and BTC. For this systematic review, a systematic review of Medline and Embase to select biomarker studies of PCa and BTC patients was conducted. Results: after reviewing 72 studies, 79 biomarker candidates were identified, including 22 nucleic acids, 43 proteins, and 14 immune cell types. Of the 72 studies, 61 examined PCa, and 11 examined BTC. Conclusion: PCa and BTC are characterized by nucleic acid, protein, and immune cell profiles that are markedly different from those of healthy subjects. These altered molecules and cell subsets may serve as cancer-specific biomarkers, particularly in blood. Further studies are needed to better understand the diagnosis and prognosis of PCa and BTC.

## 1. Introduction

Pancreatic cancer (PCa) is the fourth most common cause of death in Japan. Indeed, there were an estimated 496,000 (234,000 in Asia) new cases annually worldwide in 2020, and the number of new cases is estimated to be 802,000 (424,000 in Asia) in 2040 [1]. Despite the rapid evolution of cancer treatments in recent years, the 5-year survival rate for PCa is only 5–6% [2]. Biliary tract cancer (BTC), which includes cholangiocarcinoma (both intrahepatic cholangiocarcinoma (ICC) and extrahepatic cholangiocarcinoma (ECC)) and gallbladder cancer, also has a poor prognosis [3]. BTC has a low incidence, accounting for about 3% of all adult cancers [4]. Most cases are unresectable [3], and even if they could be found in the resectable stage, recurrence rates are very high [5,6]. The 5-year survival rate for BTC is about 5–15% [7,8]. Since PCa and BTC are less symptomatic, these cancers are generally hard to detect at an early stage. Furthermore, PCa and BTC have similar pathological characteristics, which makes it difficult to discriminate them just by blood biomarkers. Therefore, identifying good biomarkers for the diagnosis and prognosis of PCa and BTC is important for both early diagnosis and treatment. This review comprehensively summarized the current state of blood biomarker studies for PCa and BTC, which may lead to early detection and improved prognosis.

## 2. Methods

A systematic electronic search of the Medline (https://pubmed.ncbi.nlm.nih.gov, accessed on 10 February 2021) and Embase (https://www.embase.com, accessed on 10 February 2021) databases was performed to identify studies reporting the characteristics of PCa and BTC patients (Figure 1). The search was performed using the following terms with Boolean operators: (biliary OR bile OR cholangiocarcinoma OR pancreatic OR pancreas) AND biomarker AND (checkpoint OR chemoresistance) AND (blood OR serum OR plasma). The same terms were used on Medline and Embase, and duplicated articles were deleted. Articles that were not written in English, review articles, conference abstracts, and articles about cancer types other than PCa or BTC were excluded. A systematic review of articles that met the selection criteria was performed. Abstract and in-text reviews were performed by a single reviewer (H.T.). The selected research articles were cross-searched to identify additional relevant studies. This review also introduces tissue biomarkers and other biomarkers that can be applied as blood biomarkers.

## 3. Results

### 3.1. Characteristics of PCa

#### 3.1.1. Non-Coding RNA

Many circulating microRNAs were reported as biomarkers for PCa patients (Table 1). Meta-analysis revealed that, compared with healthy controls, elevated plasma miR-744 levels in PCa patients were a poor prognostic factor, contributing to reduced progression-free survival [9]. Serum miR-21 levels were also elevated and were even higher in patients with gemcitabine-resistant PCa; these elevated levels correlated with poor survival [10,11]. Serum miR-7 expression was lower in PCa patients than in controls and had an adverse effect on prognosis [12]. In addition, plasma miR-34a and miR-150 levels, and expression of miR-34a and miR-150 in tumor tissue, were lower in PCa patients than in healthy controls [13]. Plasma miR-107, miR-126, miR-451, miR-145, miR-491-5p, and miR-146b-5p levels decreased in PCa patients, with miR-107 being the most decreased miRNA [14]. PCa patients showed high amounts of miR-191, miR-21, and miR-451a in serum exosomes, and high miR-21 was associated with overall survival and resistance to chemotherapy [15]. MiR-200b and miR-200c were overexpressed in serum exosomes from PCa patients, and high expression of miR-200c in total serum exosomes and miR-200b in EpCAM-positive serum exosomes correlated with shorter overall survival (OS). In PCa patients, nucleic acids other than mRNA were also altered; for example, serum LINC01559 expression was markedly increased than in healthy controls and correlated with survival [16]. Expression of let-7 family members (especially let-7d) in the plasma of PCa patients correlated inversely with overall survival [11]. 

#### 3.1.2. Protein Expression

A very large number of protein biomarker candidates were extracted (Table 2). High expression of colon carcinoma-1 (MACC1) oncogene in serum of PCa patients correlated with lymph node metastasis, distant metastasis, and a later TNM stage [20]. Plasma IL-8 was the circulating factor that correlated most significantly with the overall survival of PCa patients [21]. PIM-1 expression was upregulated significantly in PCa tissues compared with normal tissues. In addition, plasma PIM-1 levels were significantly increased and were associated with TNM stage (II/III/IV) [22]. Although protein levels in the blood are unknown, many proteins that can be tissue biomarkers for PCa were also identified by our search algorithm. 72% of PCa patients expressed activated insulin/IGF receptors on tumor cells [23]. Expression of CD133, Notch1, Notch2, and Notch4 receptors was significantly higher in PCa tissues than in pancreatic tissues from patients with benign lesions [13]. Patients with lower levels of lactate and higher levels of human equilibrium nucleoside transporter (hENT1) in PCa tissue had better survival rates [24]. Disheveled-axin (DIX) domain (DIXDC1), a protein containing a coiled-coil domain and a DIX domain, was also highly expressed in PCa tissues and correlated with worse OS [25]. By contrast, expression of the V-domain Ig suppressor of T cell activation (VISTA) in PCa tissues was significantly associated with prolonged OS [26], and while expression of PD-1 or PD-L1 in pancreatic neuroendocrine tumors was rare, expression of PD-L2 was common in neuroendocrine tumor subtypes. Expression of immune-related proteins was also altered, with well-differentiated pancreatic neuroendocrine tumors expressing low levels of PD-1 and PD-L1 [27]. Cancer-associated pancreatic fibroblasts isolated from the tumors of PCa patients showed higher expression of PD-L1 than primary dermal fibroblasts from healthy subjects [28]. Low HLA class I expression in PCa tissues was the only risk factor for poor survival; PD-L1-negative and HLA class I high-expressing PCa was significantly associated with an increased number of infiltrating CD8+ T cells in the TME, and with improved prognosis [29]. The spindle and kinetochore-associated genes SKA1-3 were highly expressed in PCa tissues; high expression of SKA1 and SKA3 was associated with a poor prognosis [30].

#### 3.1.3. Immune Cell Types

Several biomarker candidates were also extracted for immune cells (Table 3). High CD38/CD101 co-expression by PD-1+ CD8+ T cells in the peripheral blood of PCa patients or tumor-infiltrating lymphocytes (TILs) in PCa tissues correlated significantly with tumor/node/metastasis (T/N/M) classification, and with clinical stage and survival [39]. Immune cell changes were more common in tumor tissue. Cytokines and chemokines associated with immune cells that were altered in tumor tissues may be potential biomarkers in blood. In PCa, the paraneoplastic stroma containing cancer cells harbored fewer CD8+ T cells than stroma without cancer cells [61]. By contrast, the number of tumor-infiltrating CD68+M, CD163+M2, and CD47 cells was higher, and these cells were significantly associated with decreased OS [32]. Expression of tumor antigens MYPT1, PSMC5, and TRFR was also significantly higher in PCa tissues than in healthy controls; patients with antibodies specific for these antigens showed improved disease-free survival after granulocyte macrophage colony-stimulating factor-secreting pancreatic cancer vaccine (GVAX) therapy [62]. Tumor-infiltrating T cells showed higher expression of galectin 9 than normal T cells [63]. 

### 3.2. Characteristics of BTC

#### 3.2.1. Non-Coding RNA

No candidate blood biomarkers for BTC were extracted. Cholangiocarcinoma (CCA) tissues showed lower expression of the lncRNA miR-155 host gene (miR-155HG). In addition, miR-155HG was closely associated with improved OS [19]. 

#### 3.2.2. Protein Expression

MFAP5 levels in the serum of ICC patients and expression of MFAP5 in the ECC tissues was lower than in healthy controls [59]. The co-stimulatory receptor GITR, and co-inhibitory receptors PD-1 and CTLA4, were overexpressed by TILs when compared with T cells in blood and normal tissues [52]. Some candidate biomarkers in blood that had been altered in the tissue were also extracted. In GBC tissues, elevated expression of SPTLC1 and CERS2, and that of their product C24-ceramide, was associated with tumor stage, distal metastasis, and poor prognosis [56]. Expression of BUB1B was increased in CCA tissues, and ECC patients with high expression of BUB1B showed worse OS and recurrence-free survival than those with low expression of BUB1B [58]. Cks2 was significantly elevated in BTC tissues, and its overexpression was associated with poor differentiation, CA19-9, and a poor prognosis [57]. 

#### 3.2.3. Immune Cell Types

Serum soluble FasL, MCP-1, and interferon-γ also correlated with poor prognosis in BTC patients [55]. Some immune cells that had been altered in the tissue were extracted. The percentage of cytotoxic T cells and natural killer cells was lower in CCA tissues than in normal tissues; however, the percentage of regulatory T cells was higher [52]. In ECC patients, a higher ratio of PD-1(+)/CD8(+) TILs meant lower OS, recurrence-free survival, and distant metastasis-free survival [73]. The number of PD-1+ T cells and expression of PD-L1 in the tumor tissues of ICC patients were elevated, which had a negative impact on prognosis. By contrast, high numbers of PD-1+ T cells or high expression of PD-L1 in normal tissues had no effect on prognosis [60]. Furthermore, expression of HHLA2 in ICC tissues was more common than that of PD-L1 (49.0% vs. 28.1%, respectively); overexpression of HHLA2 was associated with decreased CD3 + TIL and CD8 + TIL numbers and higher CD4+ Foxp3+/CD8 + TIL ratios, which affected OS. By contrast, infiltration of the tumor by PD-L1-expressing T cells and CD163+ tumor-associated macrophages were not associated with OS [53]. 

### 3.3. Treatment of PCa and BTC

#### 3.3.1. PCa, BTC, and Immune Checkpoint Inhibition

In recent years, immune checkpoint inhibitors have been used to treat cancer. CD8+ cytotoxic T cells are important effectors of the immune response against cancer. Immune checkpoint inhibitors are effective only when CD8+ T cells infiltrate the tumor; thus, immune checkpoint inhibitors alone are ineffective against PCa because CD8+ T cells do not infiltrate the tumor [74,75]. Many factors regulate the movement of CD8+ T cells, including activation of the tumor endothelium by T cell-derived IL-3, which triggers T cell infiltration [44], and deficiency of tumor ETS homology factor, which causes a decrease in the number of regulatory T cells, myeloid-derived suppressor cells, and tumor-infiltrating CD8+ T cells [37]. Although metastatic PCa tumors are largely resistant to anti-PD-1 therapy, blockade of PD-1 in granulin-depleted tumors restores anti-tumor immunity [36]. PD-1/PD-L1 and CD8 T cells are closely related, as PD-1 blockade can increase CD8 T cell and tumor-specific interferon-γ production in the tumor microenvironment [38] or produce anti-tumor effects by increasing KLRG1 + LAG3 - TNFα+ tumor-specific T cells in tumors [76]. Bone marrow cells inhibit CD8+ T cell anti-tumor activity by inducing expression of PD-L1 by tumor cells in an epidermal growth factor receptor (EGFR)/mitogen-activated protein kinase (MAPK)-dependent manner [67]. In addition, there are many pathways that alter immune cells, such as when gastrin is stimulated concurrently with PD-1 AB administration, tumors have less fibrosis, inhibitory Treg lymphocytes, and tumor-associated macrophages [41]. The combination of an anti-PD-L1 mouse monoclonal (MAb) and a TGF-β type I receptor small molecule kinase inhibitor (LY364947) results in the long-term survival of mice due to the influx of CD8α T cells into the TME [77]. Moreover, the combination of entinostat (ENT), a histone deacetylase inhibitor, and immune checkpoint inhibition significantly alter the infiltration and function of innate immune cells, allowing for a more potent adaptive immune response [65]. Inhibition of MLL1, a PD-L1 transcriptional activator, in combination with an anti-PD-L1 or an anti-PD-1 antibody, effectively suppresses pancreatic tumor growth in a FasL- and CTL-dependent manner [35]. In addition, immune checkpoint inhibitors are important in combination with other drugs, as inhibition of over-activated focal adhesion kinase (FAK) greatly reduces tumor fibrosis and the number of tumor-infiltrating immunosuppressive cells, making them sensitive to T-cell immunotherapy and PD-1 antagonists [33]. The role of CD8+ T cells in immune checkpoint inhibition is very important; for example, treatment with a small glutamine analog (6-diazo-5-oxo-L-norleucine [DON]) reduces the amount of hyaluronan and collagen in the TME, leading to extensive remodeling of the extracellular matrix and increased infiltration by CD8+ T cells [78]. On the other hand, other immune cells have also been implicated in anti-tumor effects, such as CD4+-dependent anti-tumor effects [66] and innate lymphocytes (ILC2), which are anticancer immune cells in PCa immunotherapy and emerge as tissue-specific enhancers of cancer immunity that amplify the efficacy of anti-PD-1 immunotherapy [64]. Partial activation of CD11b leads to repolarization of tumor-associated macrophages, reduced numbers of tumor-infiltrating immunosuppressive myeloid cells, and enhanced dendritic cell responses, all of which improve anti-tumor T cell-mediated immunity and make checkpoint inhibitors effective in previously unresponsive PCa models [34]. Targeting immune cells is important; indeed, the combination of a CD40 agonist and a PD-1 antagonist MAb exerts anti-tumor effects, which are manifested through tumor infiltration by IFNγ-, Granzyme B-, and TNFα-secreting effector T cells [79]. The combination of agonist antibodies (ABS) targeting the immunostimulatory CD40 receptor plus MEK inhibitors suppressing M2 macrophages, bone marrow-derived suppressor cells, and T regulatory cells generates a potent synergistic anti-tumor effect [69]. Mice with EHF-overexpressing tumors responded significantly better to anti-PD-1 therapy than those with control tumors [37]. In addition, anti-PD-L1 therapy sensitized pancreatic cancer cells to antiangiogenic therapy and, conversely, antiangiogenic therapy improved anti-PD-L1 therapy [80].

#### 3.3.2. Resistance of PCa and BTC to Chemotherapy

Although gemcitabine is used as a drug to treat PCa, cancer often becomes resistant; this is one of the reasons PCa is difficult to treat [81]. Cancer stem-like cells (CSLC), which can differentiate into a variety of mature cell types, have a high rate of metastasis and are implicated in resistance to chemotherapy [82]. Chemoresistant cancer cells possess stem-like characteristics, such as the ability to form spheres and high expression of cancer stem cell-like surface markers CD44/CD133 [83]. Therefore, it is important to target CSLCs. The combination of activated T cells and cutamaxomab eliminates CSLCs [46]. Soluble vascular cell adhesion molecule-1 (SVCAM-1) increases tumor resistance to gemcitabine [47]; however, combining gemcitabine with inhibition of the adhesion molecule L1CAM (CD171) reduces VEGF expression and the number of CD31-positive vessels, resulting in a stronger anti-tumor effect [45]. Cell adhesion molecules are also implicated in chemotherapy resistance. Knockdown of Slug, which regulates epithelial-mesenchymal transition (EMT), makes cells sensitive to gemcitabine [50]. Increased activity of LOX family proteins promotes chemoresistance because LOX proteins mediate collagen cross-linking and reinforce the tumor stroma and extracellular matrix (ECM), thereby promoting resistance to chemotherapy [48]. In addition, miRNAs are involved. MiR-205 reduces the expression of chemoresistance markers and re-sensitizes cells to gemcitabine [17]. In addition, the combination of ursolic acid and gemcitabine decreases Ki67 and miR-29A expression, thereby inhibiting tumor cell proliferation [84]. Expression of exosomal ephrin A receptor 2 (EphA2) may induce chemoresistance [31], whereas suppression of the NF-κB pathway may make cells sensitive to gemcitabine [85].

## 4. Discussion

There were many more studies of PCa (n = 61) than of BTC (n = 11), which may reflect the overwhelmingly larger numbers of PCa patients compared with BTC patients. However, such as PCa, BTC is a cancer that is difficult to diagnose/treat and has a poor prognosis; therefore, it is important to identify biomarkers for this cancer type. In addition, for clinical application, it is important to clarify whether PCa biomarkers can be used as BTC biomarkers in PCa and BTC, which have similar characteristics. We found differential expression of many miRNAs in PCa and BTC patients compared with healthy individuals. MiRNAs are small non-coding RNAs that negatively regulate the expression of most of the mRNAs in cells and have unique and diverse expression patterns in cancers [86]. Because miRNAs expression is altered not only in cancer tissues but also in plasma, blood-borne miRNAs such as miR-21 and miR-107 are promising biomarkers for diagnosis and prognosis of PCa and BTC. Protein expressions and immune cell compositions are also different in cancer tissues and normal tissues. Usually, tissue biomarkers are useful for predicting cancer prognosis and resistance to treatment but cannot be used for diagnosis. However, we found that many biomarkers in tissue also appeared in the blood. Thus, it is important to investigate the expression of tissue biomarkers in blood. In addition, there are many therapeutic targets for PCa and BTC, which can be useful biomarkers in blood. For example, the number of immune cells, such as cytotoxic T cells and natural killer cells, was low in PCa and BTC, whereas expression of immune checkpoint molecules, such as PD-1/PD-L1 and PD2/PDL2, was high. Immune checkpoint inhibitors are promising cancer drugs, but they are only effective when CD8+ T cells can infiltrate the tumor [74]. Hence, disease prognosis and sensitivity to immune checkpoint inhibitors can be predicted by examining immune checkpoint molecules, such as PD-1/PD-L1, and immune cells, such as CD8 + T cells, in the blood. In addition, stem cell markers may be useful biomarkers of chemotherapy resistance. Currently, cell-free DNA sequencing is in progress in the US, and miRNA testing technology is being developed in Japan. Especially, the methylation status of cell-free DNA is one of the most promising circulating biomarkers for various cancer detection [87].

## 5. Conclusions

A large number of miRNAs were extracted as candidate biomarkers in blood for the diagnosis and prognosis of PCa and BTC. A total of 37 biomarkers for diagnosis, 22 biomarkers for prognosis, and 23 biomarkers for treatment resistance were detected. Thus far, it is hard to predict which markers would be better than the others, thus that we should keep on investigating and verifying many kinds of biomarkers in parallel. Ultimately, some combinations of several different types of biomarkers will be applied to future clinical practice. Further research for biomarkers in the blood is warranted for early detection and proper management of PCa and BTC.

## Figures and Tables

**Figure 1 cancers-13-04220-f001:**
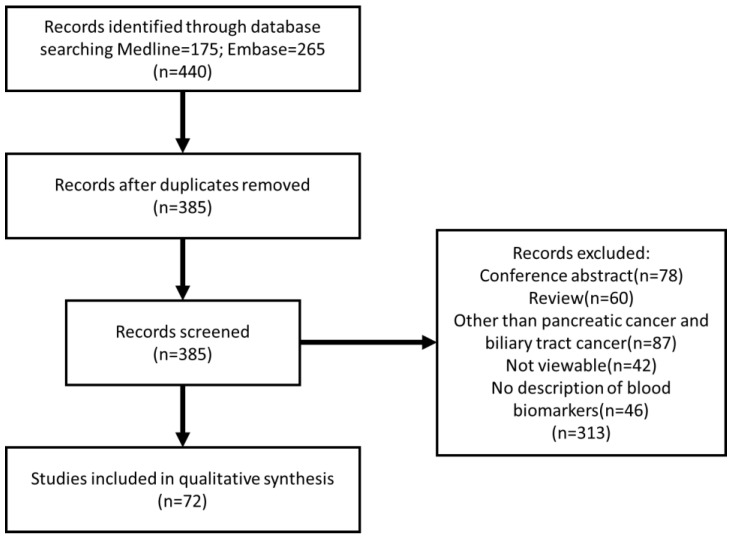
Flow diagram of literature search.

**Table 1 cancers-13-04220-t001:** Biomarker candidates for non-coding RNA identified in the present systematic review.

Cancer Type	Biomarker	Diagnosis	Prognosis	Chemoresistance	Species	Plasma	Serum	Serum Exosome	Tissue	Ref.
PCa	miR-205	X		X	Human, Mouse				Y	[17]
	miR-7	X	X		Human		Y			[12]
	miR-200b, miR-200c	X			Human			Y		[18]
	miR-191, miR-21, miR-451a	X	X		Human			Y		[15]
	miR-744	X	X		Human	Y				[9]
	LINC01559	X		X	Human		Y			[16]
	miR-34a, miR-150	X	X		Human	Y			Y	[13]
	miR-21			X	Human		Y			[10,11]
	let-7	X	X		Human	Y	Y			[11]
	miR-107	X	X		Human	Y				[14]
BTC	miR-155HG	X		X	Human				Y	[19]

PCa, pancreatic cancer; BTC, biliary tract cancer.

**Table 2 cancers-13-04220-t002:** Biomarker candidates for proteins identified in the present systematic review.

Cancer Type	Biomarker	Diagnosis	Prognosis	Chemoresistance	Species	Plasma	Serum	Tissue	Cell	Cell Exosome	Ref.
PCa	EphA2			X	Human					Y	[31]
	Galectin-9	X	X		Human			Y			[32]
	FAK	X		X	Mouse			Y			[33]
	CD11b	X		X	Human			Y			[34]
	MLL1			X	Human, Mouse			Y			[35]
	CD47		X		Human			Y			[32]
	Granulin			X	Human, Mouse			Y	Y		[36]
	EHF			X	Mouse			Y			[37]
	IFNγ			X	Human			Y			[38]
	PDL-2	X			Human			Y			[27]
	CD38/CD101	X	X		Human	Y		Y			[39]
	ATP, HMGB1			X	Mouse			Y			[40]
	Gastrin		X		Mouse			Y			[41]
	IL8	X	X		Human	Y					[21]
	PD-L1, PD-L2	X			Human			Y			[28]
	CDH3, PLAU, LFNG		X		Human	Y					[42]
	CD16	X			Human			Y			[43]
	PIM-1	X	X		Human	Y					[22]
	IL3			X	Mouse			Y			[44]
	Lactic acid	X	X		Human			Y			[24]
	CD171			X	Mouse			Y			[45]
	EpCAM, CD3			X	Human				Y		[46]
	DIXDC1	X	X		Human			Y			[25]
	sVCAM-1		X		Human, Mouse	Y		Y			[47]
	VISTA	X	X		Human			Y			[26]
	IGF	X			Human			Y			[23]
	LOX family			X	Human, Mouse			Y			[48]
	MACC1	X		X	Human		Y				[20]
	ETF			X	Mouse			Y			[37]
	IRE1α	X			Human, Mouse			Y	Y		[49]
	Slug			X	Human				Y		[50]
	ADAM family	X		X	Human			Y			[51]
	HLA class I		X		Human			Y			[29]
	SKA1, SKA3	X	X		Human			Y			[30]
BTC	GITR, CTLA4	X		X	Human			Y			[52]
	HHLA2	X	X		Human			Y			[53]
	PD-1	X		X	Human			Y			[52,54]
	FasL, MCP-1, IFNγ		X		Human		Y				[55]
	C24-Ceramide	X	X		Human			Y			[56]
	Csk2	X	X	X	Human			Y			[57]
	BUB1B		X		Human			Y			[58]
	MFAP5	X	X		Human		Y	Y			[59]
	PD-1/PD-L1	X	X		Human			Y			[60]

PCa, pancreatic cancer; BTC, biliary tract cancer.

**Table 3 cancers-13-04220-t003:** Biomarker candidates for immune cells identified in the present systematic review.

Cancer Type	Biomarker	Diagnosis	Prognosis	Chemoresistance	Species	Serum	Tissue	Cell	Ref.
PCa	ILC2			X	Human, Mouse		Y	Y	[64]
	CD8(+) T cells	X		X	Human, Mouse		Y		[61,65,66]
	Myeloid cells	X			Mouse		Y		[67]
	Mesothelin-specific cells		X		Human	Y			[68]
	M2-type macrophages			X	Mouse		Y		[69]
	MYPT1, PSMC5, TRFR	X			Human		Y		[62]
	TAM		X		Mouse		Y		[70]
	CD4+ T cells			X	Mouse		Y		[66]
	Treg	X			Human		Y		[71]
	T cells, NK cells		X		Mouse		Y		[72]
BTC	PD-1(+)/CD8(+) TILs		X		Human		Y		[73]

PCa, pancreatic cancer.

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
