# Peer review of "Challenges for Better Diagnosis and Management of Pancreatic and Biliary Tract Cancers Focusing on Blood Biomarkers: A Systematic Review"

_cancers, 2021, doi:10.3390/cancers13164220_

Round 1

Reviewer 1 Report

Dear authors,

Many thanks for reviewing PC and BTC biomarkers, very useful.

I have comments on microRNAs as biomarkers, in particular when microRNA quantity is lower in PC patients. These cannot be originally enriched in pancreas, right? or they are absorbed somehow...How do you explain that? Are they real, artefacts or perhaps blood cell subtype biomarkers (i.e. CD8T cells) instead? Can you comment and clarify please as I'm curious to hear your thoughts about that?

 I have another comment regarding the semantic. You mentioned up- and down-regulated microRNA in plasma/serum. As far as I am aware, these microRNAs are not regulated at the transcriptional level but rather detected as a consequence of tissue damaging or of the abundance of cell subset. I suggest you modify the wording accordingly.

At last, you refer microRNA, protein and immune cells as diagnostic and prognostic biomarkers of PC and BTC. I understand well that all these biomarkers are likely good to diagnose PC and BTC mainly at late stage. I am more suspicious on their prognostic potential. Can you comment on that please? What about early biomarkers? do you have suggestions or ideas on where to search and how to characterize these?  

Kind regards

Author Response

We are grateful to the reviewers for their critical comments and insightful suggestions, which have helped us to considerably improve our paper. As indicated in the responses that follow, we have taken these comments and suggestions into account in the revised version of our paper. We have changed the text in the manuscript to accommodate these comments.

Comment: 1. Line 22: Remove “Materials and Methods”. Rephrase using “For this systematic review, …”

Reply: Thank you for your suggestion. We've changed the text as follows (lines 22):

“For this systemtic review, A systematic review of Medline and Embase to select biomarker studies of PCa and BTC patients was conducted.”

Comment: 2. Line 34-35: GLOBCAN numbers given but not clear if these are really total or nationwide. One distinct feature of these cancers is that the incident rates are going up. Using the same interactive GLOBCAN source, please pull out the current (2020) worldwide and Asian numbers, and the GLOBCAN prediction for year 2040. As these are somewhat shocking, they will emphasise the importance of your own review.

Reply: We apologize for our unclear description. This sentence was revised as follows (lines 34-35):

Old: “Indeed, an estimated 460,000 new cases occur annually nationwide, of which 430,000 are expected to be fatal [1].”

New: “Indeed, there were an estimated 496,000 (234,000 in Asia) new cases annually worldwide in 2020, and the number of new cases is estimated to be 802,000 (424,000 in Asia) in 2040[1].”

Comment: 3. Line 38: The reference 3 is quite outdated and limited so use this instead: 3. Jansen H, Pape UF, Utku N. A review of systemic therapy in biliary tract carcinoma. J Gastrointest Oncol. 2020 Aug;11(4):770-789. doi: 10.21037/jgo-20-203. PMID: 32953160; PMCID: PMC7475338.

Reply: Thank you for your great suggestion. Accordingly, we changed the reference 3.

Comment: 4. Line 47: Although Medline is commonly used, Embase is less familiar. Hence, it might be worth to give the source (i.e. weblink) for both, especially as there is a company MEDLINE appearing first in Google searches – this medical company has nothing to do with literature searches.

Reply: Thank you for kind suggestion. We added weblink as follows: “A systematic electronic search of the Medline (https://pubmed.ncbi.nlm.nih.gov) and Embase (https://www.embase.com) databases was performed” (lines 50-51).

Comment: 5. Line 49: Apparently you performed the filtering using MEDLINEs Boolean operators (AND/OR etc.)? If so, this could be mentioned in the text, as well if you used the same approach in Embase, and how did you combine the filtered data further (Excel ?).

Reply: We apologize for our unclear description. We performed the filtering using Boolean operators (AND/OR) both in Medline and Embase. Then every duplicated articles were deleted based on PMID using Excel. This point was clarified as follows: (line 52-56)

“The search was performed using the following terms with Boolean operators: (biliary OR bile OR cholangiocarcinoma OR pancreatic OR pancreas) AND biomarker AND (check-point OR chemoresistance) AND (blood OR serum OR plasma). The same terms were used on the Medline and Embase, and duplicated articles were deleted.”

Comment: 6. Line 61-62: The in-text citation for Table 1 is here (page 2), but Table 1 itself is on page 6, which makes it very difficult for the reader to follow the text. The Table 1 should be located soon after that in-text citation. The same applies to ALL tables in this review.

Reply: Thank you for your comment. I changed the position of the Tables 1, 2, and 3.

Comment: 7. Line 237, 246 and 252 have used “therefore”, at least one should be replaced with “hence” etc. As a general note, many of these studies are “fishing expeditions”, i.e using qPCR and one biomarker for testing and would need further verification or use of systematic approach, like NGS has been used for BTC mutations. Hence, it should be mentioned that mutational biomarkers have been excluded from this review, “for systematic analysis see (ref)”, for the readers to understand the scope of your review:

Reply: Thank you for your comment. We replaced with “thus” and “hence” (lines 328, 334). We also mentioned about the cell-free DNA sequencing with proper citation in the discussion section (lines 338-40) although this article was not picked up by our search strategy because the corresponding article is not specifically described about pancreatic/bile duct cancer.

“Currently, cell-free DNA sequencing is in progress in the US, and miRNA testing technology is being developed in Japan. Especially, the methylation status of cell-free DNA is one of the most promising circulating biomarkers for various cancer detection [87].”

.

Comment: 8. This part is way too short as it is only two sentences. At minimum, the numbers of unique biomarkers found for chemoresistance, diagnosis and prognosis should be mentioned here.

Reply: Thank you for your suggestion. We revised the conclusion section as follows: “A large number of miRNAs were extracted as candidate biomarkers in blood for the diagnosis and prognosis of PCa and BTC. Thirty-seven biomarkers for diagnosis, 22 bi-omarkers for prognosis, and 23 biomarkers for treatment resistance were detected. So far, it is hard to predict which markers would be better than the others, so that we should keep on investigating and verifying many kinds of biomarkers in parallel. Ultimately, some combinations of several different types of biomarkers will be applied to future clinical practice. Further researches for biomarkers in blood are warranted for early detection and proper management of PCa and BTC.”

Comment: 9. As mentioned before the tables should be in close proximity where they have been mentioned in the text. The tables are also quite hard to read so I would suggest using columns and “X” marks when applicable. For example, Table 1 has under “Evaluated item” three types: “Diagnosis”, “Chemoresistance” and “Prognosis”. With three vertical columns, one for each item, you can use simple tick (“X”) marks to indicate if applicable for that biomarker. So for example, mir-200b, mir-200c would have only tick mark in “Diagnosis” column but miR-1 in both “Diagnosis” and “Prognosis”. This would allow the reader skim through the table for the biomarkers of certain type. The same approach can be used for “Source”. The same model can be used for Table 2 and Table 3 as well.

In some cases, you have the same biomarker indicated by two referred literature sources. You could combine these in one line, as this would help the readability. For example. Table 3 you could combine CD8+ T cells and give reference as [29, 53, 57].

Reply: Thank you for your suggestion. We changed the Tables 1, 2, and 3.

Reviewer 2 Report

Abstract

Line 22: Remove “Materials and Methods”. Rephrase using “For this systematic review, …”

Introduction

Line 34-35: GLOBCAN numbers given but not clear if these are really total or nationwide. One distinct feature of these cancers is that the incident rates are going up. Using the same interactive GLOBCAN source, please pull out the current (2020) worldwide and Asian numbers, and the GLOBCAN prediction for year 2040. As these are somewhat shocking, they will emphasise the importance of your own review.

Line 38: The reference 3 is quite outdated and limited so use this instead: 3. Jansen H, Pape UF, Utku N. A review of systemic therapy in biliary tract carcinoma. J Gastrointest Oncol. 2020 Aug;11(4):770-789. doi: 10.21037/jgo-20-203. PMID: 32953160; PMCID: PMC7475338.

Methods

Line 47: Although Medline is commonly used, Embase is less familiar. Hence, it might be worth to give the source (i.e. weblink) for both, especially as there is a company MEDLINE appearing first in Google searches – this medical company has nothing to do with literature searches.

Line 49: Apparently you performed the filtering using MEDLINEs Boolean operators (AND/OR etc.)? If so, this could be mentioned in the text, as well if you used the same approach in Embase, and how did you combine the filtered data further (Excel ?).

Results

Line 61-62: The in-text citation for Table 1 is here (page 2), but Table 1 itself is on page 6, which makes it very difficult for the reader to follow the text. The Table 1 should be located soon after that in-text citation. The same applies to ALL tables in this review.

Discussion

Line 237, 246 and 252 have used “therefore”, at least one should be replaced with “hence” etc. As a general note, many of these studies are “fishing expeditions”, i.e using qPCR and one biomarker for testing and would need further verification or use of systematic approach, like NGS has been used for BTC mutations. Hence, it should be mentioned that mutational biomarkers have been excluded from this review, “for systematic analysis see (ref)”, for the readers to understand the scope of your review:

Ahn DH, Javle M, Ahn CW, Jain A, Mikhail S, Noonan AM, Ciombor K, Wu C, Shroff RT, Chen JL, Bekaii-Saab T. Next-generation sequencing survey of biliary tract cancer reveals the association between tumor somatic variants and chemotherapy resistance. Cancer. 2016 Dec 1;122(23):3657-3666. doi: 10.1002/cncr.30247. Epub 2016 Aug 6. Erratum in: Cancer. 2017 Jun 15;123(12 ):2376. PMID: 27495988; PMCID: PMC5222890.

Conclusions

This part is way too short as it is only two sentences. At minimum, the numbers of unique biomarkers found for chemoresistance, diagnosis and prognosis should be mentioned here.

Tables

As mentioned before the tables should be in close proximity where they have been mentioned in the text. The tables are also quite hard to read so I would suggest using columns and “X” marks when applicable. For example, Table 1 has under “Evaluated item” three types: “Diagnosis”, “Chemoresistance” and “Prognosis”. With three vertical columns, one for each item, you can use simple tick (“X”) marks to indicate if applicable for that biomarker. So for example, mir-200b, mir-200c would have only tick mark in “Diagnosis” column but miR-1 in both “Diagnosis” and “Prognosis”. This would allow the reader skim through the table for the biomarkers of certain type. The same approach can be used for “Source”. The same model can be used for Table 2 and Table 3 as well.

In some cases, you have the same biomarker indicated by two referred literature sources. You could combine these in one line, as this would help the readability. For example. Table 3 you could combine CD8+ T cells and give reference as [29, 53, 57].

Author Response

We are grateful to the reviewers for their critical comments and insightful suggestions, which have helped us to considerably improve our paper. As indicated in the responses that follow, we have taken these comments and suggestions into account in the revised version of our paper. We have changed the text in the manuscript to accommodate these comments.

Comment: Well written article, pleasure and informative to read

Reply: We fully appreciate your kind comment.

Reviewer 3 Report

Well written article, pleasure and informative to read

Author Response

We are grateful to the reviewers for their critical comments and insightful suggestions, which have helped us to considerably improve our paper. As indicated in the responses that follow, we have taken these comments and suggestions into account in the revised version of our paper. We have changed the text in the manuscript to accommodate these comments.

Comment: 1. In the Abstract / Results, authors define the number of studies and detected markers but they refer to them as one group. However throughout the article PC and BTC are divided in separate groups. From this perspective, the cumulative numbers in the Abstract don´t represent the results in the article. In the Introduction there is a short explanation needed why the authors selected these 2 cancer types for the review, what is the connection between PC and BTC?

Reply: Thank you for your important comment. We added the sentence as follows (lines 42-53): “Since PCa and BTC are less symptomatic, these cancers are generally hard to detect at an early stage. Furthermore, PCa and BTC have similar pathological characteristics, which makes it difficult to discriminate them just by blood biomarkers”.

Comment: 2. The title is "Blood biomarkers...." However only a minority of the results describes the expression of potential markers in the blood. Most of the text describes the protein expression in the tissues and treatment options. Maybe the title could be slightly rephrased to more specifically represent the article content.

Reply: Thank you for your comment. Our search strategy was designed to systematically review blood biomarkers but in fact, we also described tissue expression profiles to help readers’ better understanding. We changed the title to “Challenges for better diagnosis and management of pancreatic and biliary tract cancers focusing on blood biomarkers: a systematic review”. We also added in the text (lines 68-70): “This review also introduces tissue biomarkers and other biomarkers that can be applied as blood biomarkers”.

Comment: 3. In DIscussion authors are stating that miRNAs degrade mRNA which is not the only mechanism how miRNAs operate. This part should be slightly expanded and better explained why is this important in the given context.

Reply: We apologize for our unclear description. This sentence was revised as follows (lines 319-321):

Old: “MiRNAs are non-coding RNAs that regulate gene expression by degrading messenger RNA and inhibiting its translation.”

New: “MiRNAs are small non-coding RNAs that negatively regulate the expression of most of the mRNAs in cells and have unique and diverse expression patterns in cancers [86].”

Comment: 4. In line 240 it says "...miRNAs are altered.." I assume the authors meant that it is miRNAs expression that is altered not miRNA itself.

Reply: Thank you for your suggestion. We changed the text as follows (lines 322): “Because miRNAs expression is altered not only in cancer tissues but also in plasma, blood-borne miRNAs such as miR-21 and miR-107 are promising biomarkers for diagnosis and prognosis of PCa and BTC. Protein expressions and immune cell compositions are also different in cancer tissues and normal tissues.”.

Comment: 5. The Conclusion paragraph could give more specific overview of the findings. Is there sufficient results available for any relevant conclusion potentially interesting for the clinical application? Which type of marker seems the most suitable according to the currently available literature? Which type of studies are needed to translate the research into the patient care? etc.

Reply: Thank you for your suggestion. We revised the conclusion section as follows: “A large number of miRNAs were extracted as candidate biomarkers in blood for the diagnosis and prognosis of PCa and BTC. Thirty-seven biomarkers for diagnosis, 22 bi-omarkers for prognosis, and 23 biomarkers for treatment resistance were detected. So far, it is hard to predict which markers would be better than the others, so that we should keep on investigating and verifying many kinds of biomarkers in parallel. Ultimately, some combinations of several different types of biomarkers will be applied to future clinical practice. Further researches for biomarkers in blood are warranted for early detection and proper management of PCa and BTC.”

Reviewer 4 Report

In the Abstract / Results, authors define the number of studies and detected markers but they refer to them as one group. However throughout the article PC and BTC are divided in separate groups. From this perspective, the cumulative numbers in the Abstract don´t represent the results in the article. In the Introduction there is a short explanation needed why the authors selected these 2 cancer types for the review, what is the connection between PC and BTC? The title is "Blood biomarkers...." However only a minority of the results describes the expression of potential markers in the blood. Most of the text describes the protein expression in the tissues and treatment options. Maybe the title could be slightly rephrased to more specifically represent the article content. In DIscussion authors are stating that miRNAs degrade mRNA which is not the only mechanism how miRNAs operate. This part should be slightly expanded and better explained why is this important in the given context. In line 240 it says "...miRNAs are altered.." I assume the authors meant that it is miRNAs expression that is altered not miRNA itself. The Conclusion paragraph could give more specific overview of the findings. Is there sufficient results available for any relevant conclusion potentially interesting for the clinical application? Which type of marker seems the most suitable according to the currently available literature? Which type of studies are needed to translate the research into the patient care? etc.

Author Response

Comment: 1. In the Abstract / Results, authors define the number of studies and detected markers but they refer to them as one group. However throughout the article PC and BTC are divided in separate groups. From this perspective, the cumulative numbers in the Abstract don´t represent the results in the article. In the Introduction there is a short explanation needed why the authors selected these 2 cancer types for the review, what is the connection between PC and BTC?

Reply: Thank you for your important comment. We added the sentence as follows (lines 42-53): “Since PCa and BTC are less symptomatic, these cancers are generally hard to detect at an early stage. Furthermore, PCa and BTC have similar pathological characteristics, which makes it difficult to discriminate them just by blood biomarkers”.

Comment: 2. The title is "Blood biomarkers...." However only a minority of the results describes the expression of potential markers in the blood. Most of the text describes the protein expression in the tissues and treatment options. Maybe the title could be slightly rephrased to more specifically represent the article content.

Reply: Thank you for your comment. Our search strategy was designed to systematically review blood biomarkers but in fact, we also described tissue expression profiles to help readers’ better understanding. We changed the title to “Challenges for better diagnosis and management of pancreatic and biliary tract cancers focusing on blood biomarkers: a systematic review”. We also added in the text (lines 68-70): “This review also introduces tissue biomarkers and other biomarkers that can be applied as blood biomarkers”.

Comment: 3. In DIscussion authors are stating that miRNAs degrade mRNA which is not the only mechanism how miRNAs operate. This part should be slightly expanded and better explained why is this important in the given context.

Reply: We apologize for our unclear description. This sentence was revised as follows (lines 319-321):

Old: “MiRNAs are non-coding RNAs that regulate gene expression by degrading messenger RNA and inhibiting its translation.”

New: “MiRNAs are small non-coding RNAs that negatively regulate the expression of most of the mRNAs in cells and have unique and diverse expression patterns in cancers [86].”

Comment: 4. In line 240 it says "...miRNAs are altered.." I assume the authors meant that it is miRNAs expression that is altered not miRNA itself.

Reply: Thank you for your suggestion. We changed the text as follows (lines 322): “Because miRNAs expression is altered not only in cancer tissues but also in plasma, blood-borne miRNAs such as miR-21 and miR-107 are promising biomarkers for diagnosis and prognosis of PCa and BTC. Protein expressions and immune cell compositions are also different in cancer tissues and normal tissues.”.

Comment: 5. The Conclusion paragraph could give more specific overview of the findings. Is there sufficient results available for any relevant conclusion potentially interesting for the clinical application? Which type of marker seems the most suitable according to the currently available literature? Which type of studies are needed to translate the research into the patient care? etc.

Reply: Thank you for your suggestion. We revised the conclusion section as follows: “A large number of miRNAs were extracted as candidate biomarkers in blood for the diagnosis and prognosis of PCa and BTC. Thirty-seven biomarkers for diagnosis, 22 bi-omarkers for prognosis, and 23 biomarkers for treatment resistance were detected. So far, it is hard to predict which markers would be better than the others, so that we should keep on investigating and verifying many kinds of biomarkers in parallel. Ultimately, some combinations of several different types of biomarkers will be applied to future clinical practice. Further researches for biomarkers in blood are warranted for early detection and proper management of PCa and BTC.”